

# Brief communication: Wind inflow observation from load harmonics – wind tunnel validation of the rotationally symmetric formulation

**Marta Bertelè[1], Carlo L. Bottasso[1], and Stefano Cacciola[2]**

[1]Wind Energy Institute, Technische Universität München, Garching bei München 85748, Germany
[2]Dipartimento di Scienze e Tecnologie Aerospaziali, Politecnico di Milano, Milano 20156, Italy

**Correspondence:** Carlo L. Bottasso (carlo.bottasso@tum.de)

**Abstract.** TS1 The present paper further develops and experimentally validates the previously published idea of estimating the wind inflow at a turbine rotor disk from the machine response. A linear model is formulated that relates one per revolution (1P) harmonics of the in- and out-of-plane blade root bending moments to four wind parameters, representing vertical and horizontal shears and misalignment angles. Improving on this concept, the present work exploits the rotationally symmetric behavior of the rotor in the formulation of the load-wind model. In a nutshell, this means that the effects on the loads of the vertical shear and misalignment are the same as those of the horizontal quantities, simply shifted by $\pi/2$. This results in a simpler identification of the model, which needs a reduced set of observations. The performance of the proposed method is first tested in a simulation environment and then validated with an experimental data set obtained with an aeroelastically scaled turbine model in a boundary layer wind tunnel.

## 1 Introduction

The ability to control a system is often intimately linked to the awareness of the surrounding environment. For a wind turbine, the environment is represented by the wind inflow, which is characterized by speed, direction, shears, veer, turbulence intensity, presence of impinging wakes, etc. Such parameters have a profound effect on the response of a single wind turbine as well as on clusters of interacting machines within a power plant. Better awareness of the wind environment can be translated into better turbine-level and plant-level operation and control.

The current standard equipment mounted on board wind turbines for the measurement of the wind inflow is composed of one or more anemometers and wind vanes, typically located at hub height, either on the nacelle or on the spinner. Even when properly calibrated, all such devices suffer from one inherent unavoidable limitation: they provide measurements at the single point in space where they are located. As such, they are necessarily blind to all wind characteristics

that imply wind variations across the rotor disk. Alternative sensors are represented by lidars, which are, however, not yet routinely installed on board wind turbines because of cost, availability, reliability, effects due to weather conditions and lifetime issues. In this sense, current wind turbines have only a very limited awareness of the environment in which they operate.

The concept of the "rotor as a sensor" was developed to address the limitations of current wind measurement devices. The idea is conceptually very simple: changes in the wind inflow produce changes in the wind turbine response. If the wind-response map is known, one can then measure the response (for example, in the form of loads and/or accelerations) and estimate the inflow by inverting the map.

Various formulations have been proposed for this concept (Bottasso et al., 2010; Bottasso and Riboldi, 2014; Simley and Pao, 2014; Bottasso and Riboldi, 2015). In this paper we improve on the work described by Cacciola et al. (2016a) and Bertelè et al. (2017, 2018) TS2. The approach parameterizes the inflow in terms of four quantities: vertical and horizon-

tal shears and misalignment angles. The wind-response map relates these four wind states to the 1P in- and out-of-plane blade root bending moments. Both linear and quadratic maps were considered in Bertelè et al. (2017), with a marginally better accuracy for the latter. System identification was used to find the model coefficients from simulations performed with an aeroservoelastic model in a variety of wind conditions, spanning the range of interest of the four wind states. Results indicate a better accuracy of the shears than the angles, although the latter are still well captured in their mean values.

Despite the more than promising results reported in Bertelè et al. (2017), the identification of the model relating wind states to load harmonics can be cumbersome. In fact, a data set is required that covers a desired range of the four wind states. While this is not a major issue in a simulation environment where one can generate all desired wind conditions, an identification based on field test data might not be easy or even possible. In fact, some wind parameters might not change much at a given site, e.g. upflow angle and horizontal shear. This would clearly be a major hurdle, as a model only knows what is in the data used for training it.

To address this issue, the present work exploits the rotationally symmetric behavior of the rotor. In fact, the effect caused by a horizontal shear on the rotor response is the same as that caused by a vertical shear, only shifted by $\pi/2$. Similarly, the effect of a vertical upflow angle is the same of a horizontal yaw misalignment, again shifted by $\pi/2$. This means that one can collect data sets containing the desired changes in vertical shears and yaw misalignments, and identify a model that is also capable of representing the same range of horizontal shears and upflow angles.

The paper is organized as follows. Section 2 first introduces the wind parameterization and the wind-load map, and then uses the rotational symmetry of the rotor to eliminate some of the model coefficients from the identification problem unknowns. Section 3 compares the results of the new formulation to the original one first by simulations – conducted with an aeroservoelastic model – and then experimentally – using a scaled turbine in a wind tunnel. Finally, the work is closed by Sect. 4, where conclusions are drawn.

## 2 Formulation

### 2.1 Wind parameterization and rotational symmetry

The wind inflow is characterized in terms of four so-called wind states, which are defined as the vertical (upflow) and horizontal (yaw) misalignment angles $\chi$ and $\phi$, respectively, and the vertical and horizontal linear shears $\kappa_\mathrm{v}$ and $\kappa_\mathrm{h}$, respectively. These quantities should be regarded as rotor-equivalent fits of the actual spatial distribution of the wind impinging on the rotor disk at a certain instant of time.

The wind states are defined with respect to a nacelle-attached reference frame $(\boldsymbol{x}, \boldsymbol{y}, \boldsymbol{z})$ TS3 centered at the hub as

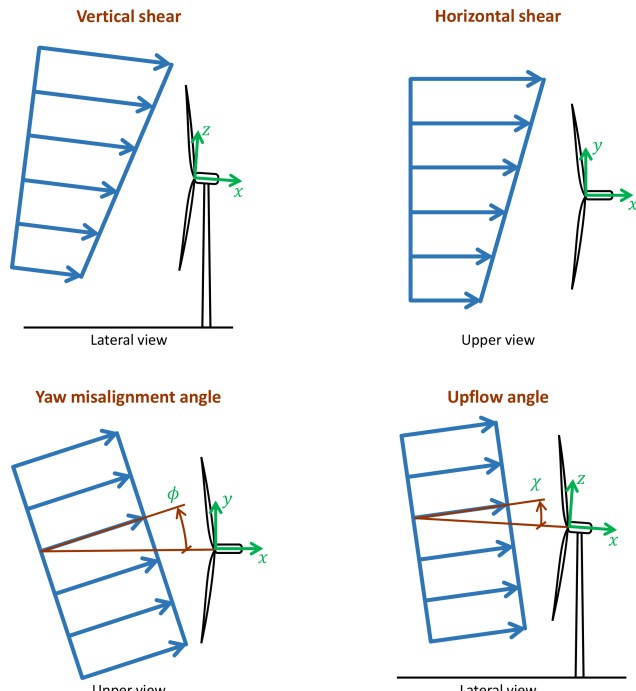

**Figure 1.** TS5 Definition of the four wind states used for parameterizing the wind field over the rotor disk.

shown in Fig. 1: unit vector $\boldsymbol{x}$ is aligned with the rotor axis and faces downwind, $\boldsymbol{z}$ points upward in the vertical plane, while $\boldsymbol{y}$ is defined according to the right-hand rule. The components of the wind vector in the nacelle-attached frame of reference are noted $\boldsymbol{V} = \{u, v, w\}^\mathrm{T}$ TS4 and they write

$$u(y, z) = W(y, z)\cos(\phi)\cos(\chi), \tag{1a}$$
$$v(y, z) = W(y, z)\sin(\phi)\cos(\chi), \tag{1b}$$
$$w(y, z) = W(y, z)\sin(\chi), \tag{1c}$$

where $W(y, z)$ is a linearly sheared wind field

$$W(y, z) = V_\mathrm{H}\left(1 + \frac{z}{R}\kappa_\mathrm{v} + \frac{y}{R}\kappa_\mathrm{h}\right), \tag{2}$$

$V_\mathrm{H}$ being the wind speed at hub height, and $R$ the rotor radius. According to this definition, the yaw misalignment and upflow angles are positive when the wind blows from the left and the lower part of the rotor, respectively, when looking upstream.

Notice that the formulation of Cacciola et al. (2016a) used a horizontal reference frame with respect to the terrain, while in the present case the frame is aligned with the rotor axis. Together with the assumed linearity of both shears, this is necessary in order to exploit the rotational symmetry of the rotor response. Hence, if the rotor is uptilted, one will have to transform the nacelle-frame wind components into a frame aligned with the ground if necessary.

Looking at Eq. (2), it appears that the effect of the vertical shear $\kappa_\mathrm{v}$ on the velocity distribution is the same of the one

caused by the horizontal shear $\kappa_h$, when rotated by $\pi/2$. On the other hand, looking at Eq. (1a–c TS6), the effect of the angles $\phi$ and $\chi$ is more complex. To eliminate this problem, the rotor-in-plane wind velocity components can be expressed in terms of the new variables

$$\widetilde{v} = \frac{v(0,0)}{V_H} = \sin(\phi)\cos(\chi), \tag{3a}$$

$$\widetilde{w} = \frac{w(0,0)}{V_H} = \sin(\chi), \tag{3b}$$

which, respectively, represent the nondimensional horizontal and vertical wind cross flows at the hub. This change in variables results in

$$u(y,z) = W(y,z)\sqrt{1 - \widetilde{v}^2 - \widetilde{w}^2}, \tag{4a}$$

$$v(y,z) = W(y,z)\widetilde{v}, \tag{4b}$$

$$w(y,z) = W(y,z)\widetilde{w}. \tag{4c}$$

With this reformulation, the effect of $\widetilde{v}$ on $v$ is the same as the effect of $\widetilde{w}$ on $w$, when rotated by $\pi/2$. Given $\widetilde{v}$ and $\widetilde{w}$, the misalignment angle $\phi$ and upflow $\chi$ can be readily recovered by inverting their respective definitions (Eq. 3a–b TS7):

$$\chi = \arcsin(\widetilde{w}), \tag{5a}$$

$$\phi = \arcsin(\widetilde{v}/\cos\chi), \tag{5b}$$

although for small angles the difference between the two sets of variables will be negligible.

## 2.2 Wind observer formulation

In this work, the linear model of Cacciola et al. (2016a) and Bertelè et al. (2017) is used to relate inflow conditions and machine response. The model writes

$$\begin{aligned} m &= \mathbf{F}(V,\varrho)\theta + m_0(V,\varrho) \\ &= [\mathbf{F}(V,\varrho)\ m_0(V,\varrho)]\begin{bmatrix} \theta \\ 1 \end{bmatrix} = \mathbf{T}\,\bar{\theta}, \end{aligned} \tag{6}$$

where $m$ is the load vector, $\theta = \{\widetilde{v}\ \kappa_v\ \widetilde{w}\ \kappa_h\}^T$ is the wind state vector, while $\mathbf{F}$ and $m_0$ represent the model coefficients, scheduled with respect to wind speed $V$ and air density $\varrho$. The load vector is defined as

$$m = \left\{ m_{1c}^{OP},\ m_{1s}^{OP},\ m_{1c}^{IP},\ m_{1s}^{IP} \right\}^T, \tag{7}$$

where $m$ indicates the blade bending moment, subscripts $(\cdot)_{1s}$ and $(\cdot)_{1c}$ TS8, respectively, indicate sine and cosine harmonics, while superscripts $(\cdot)^{OP}$ and $(\cdot)^{IP}$, respectively, out- and in-plane components. The load harmonics are readily computed via the Coleman and Feingold transformation (Coleman and Feingold, 1958) once three measured blade loads are available. For simplicity and brevity, the present paper only considers a linear wind-response map. However, non-linearities in the map can be readily included, as shown by Bertelè et al. (2017).

To identify the model coefficients $\mathbf{T}$, one should collect a rich enough data set for which both wind states $\theta$ and associated blade loads $m$ are known. Stacking side by side the $i$th wind and load vectors into matrices $\Theta$ and $\mathbf{M}$, one gets

$$\mathbf{M} = \mathbf{T}\Theta. \tag{8}$$

Finally, the model coefficients are readily identified as

$$\mathbf{T} = \mathbf{M}\Theta^T(\Theta\Theta^T)^{-1}. \tag{9}$$

The invertibility of the system is discussed in Bertelè et al. (2017).

Once the model expressed by Eq. (6) has been identified, it can be used to express the dependency of given measured loads $m_M$ on the wind states,

$$m_M = \mathbf{F}\theta + m_0 + r, \tag{10}$$

where $r$ is the measurement error, and the dependency on $V$ and $\varrho$ has been dropped for a simpler notation. The least squares estimate of the wind states $\theta_E$ is then readily obtained as

$$\theta_E = \left(\mathbf{F}^T\mathbf{R}^{-1}\mathbf{F}\right)^{-1}\mathbf{F}^T\mathbf{R}^{-1}(m_M - m_0), \tag{11}$$

where $\mathbf{R} = \mathbf{E}[rr^T]$ is the covariance weighting matrix. Given $\theta_E$, the misalignment and upflow angles can be recovered by using Eq. (5).

## 2.3 Rotational symmetry

By considering the rotational symmetry of the rotor, the number of unknown coefficients in $\mathbf{F}$ can be reduced. Indeed, a vertical shear will cause the same response of an equivalent horizontal shear, simply shifted by an azimuthal delay of $\pi/2$. The same consideration holds for the vertical and horizontal cross flows. This rotational symmetry is reflected in the derivatives of the loads with respect to the wind states, i.e., in the coefficients of matrix $\mathbf{F}$. By a rotation of $\pi/2$, the load component $m_{1c}$ becomes $m_{1s}$, while the load component $m_{1s}$ becomes $-m_{1c}$. As a result, the following conditions apply between pairs of model coefficients:

$$\frac{\partial m_{1c}}{\partial \widetilde{v}} = \frac{\partial m_{1s}}{\partial \widetilde{w}}, \tag{12a}$$

$$\frac{\partial m_{1s}}{\partial \widetilde{v}} = -\frac{\partial m_{1c}}{\partial \widetilde{w}}, \tag{12b}$$

$$\frac{\partial m_{1c}}{\partial \kappa_h} = \frac{\partial m_{1s}}{\partial \kappa_v}, \tag{12c}$$

$$\frac{\partial m_{1s}}{\partial \kappa_h} = -\frac{\partial m_{1c}}{\partial \kappa_v}. \tag{12d}$$

These conditions apply to both the out- and the in-plane components.

The term $\boldsymbol{m}_0$ in Eq. (6) represents the effects of gravity on the loads (Bertelè et al., 2017). Since this term is nonsymmetric, no reduction of these coefficients is possible in this case.

The advantage of this approach is not only in the reduced number of unknown model coefficients, but, most importantly, in the reduced datapoints necessary for identification. In fact, by eliminating the coefficients of horizontal shear and upflow angle, one can use tests in which only yaw misalignment angle and vertical shear are changing. Therefore, since the model is linear and depends on two parameters, a minimum of only three operating conditions is required for identification.

## 3 Results

### 3.1 Verification in a simulation environment

The proposed method was first tested by numerical simulations, using the model of a horizontal-axis three-bladed 3 MW wind turbine. The machine has a rotor diameter of 93 m; a hub height of 80 m; 4.5° TS9 of nacelle uptilt; and cut-in, rated and cut-out speeds equal to 3, 12.5 and 25 m s$^{-1}$, respectively. A transition region $II\,1/2$ TS10 connects the partial- and full-load regimes, extending between 9 and 12.5 m s$^{-1}$. The machine response was simulated by the aeroservoelastic multibody software Cp-Lambda (Bauchau et al., 2003; Bottasso and Croce, 2006), which is based on a geometrically exact finite element formulation. The model includes flexible blades, tower and drive train, and compliant foundations. The collective pitch and torque controller is implemented according to Riboldi (2012) and Bottasso et al. (2012), while generator and pitch actuators are modeled as first- and second-order dynamical systems, respectively. The aerodynamic rotor model is based on blade element momentum theory (BEM), augmented by classical tip and root losses, unsteady aerodynamics and dynamic stall models. Turbulent wind time histories were generated with the TurbSim code (Jonkman and Kilcher, 2012) in accordance with the Kaimal model, at the nodes of a square grid overlapping the rotor disk. "Ground truth" values of the wind states – to be used for assessing the quality of observed quantities – were obtained by fitting the instantaneous wind field at the grid nodes to the rotor swept area.

Turbulent simulations were run for a duration of 10 min, according to standard practice. The 1P harmonics were computed by the Coleman and Feingold transformation (Coleman and Feingold, 1958), using in- and out-of-plane bending moment components measured by strain gauges placed at the root of each blade. The resulting signal was finally cleaned with a low-pass filter; on-line adaption of the filter parameters was used to account for changes in rotational speed due to turbulent wind fluctuations.

Two observation models were identified. The first is the linear formulation of Bertelè et al. (2017), which does not exploit the rotational symmetry of the rotor, while the second is the linear rotationally symmetric formulation of the present paper. In the first case, the model was identified from nonturbulent wind cases corresponding to all combinations of the following wind parameters:

$$\phi = [0 \quad 16]°, \tag{13a}$$
$$\kappa_v = [0.06 \quad 0.18], \tag{13b}$$
$$\chi = [4.5 \quad 16.5]°, \tag{13c}$$
$$\kappa_h = [0 \quad -0.1]. \tag{13d}$$

TS11 A separate identification was performed for each wind speed, considering the values $V = [3\ 4\ 5\ 6\ 7\ 8\ 9\ 11\ 15\ 19]\,\mathrm{m\,s^{-1}}$. A second model was obtained by exploiting symmetry and linearity. Accordingly, the identification set was reduced to the following wind parameter combinations:

$$\phi = [0\ 16]°, \tag{14a}$$
$$\kappa_v = [0.06 \quad 0.18], \tag{14b}$$
$$\chi = 4.5°, \tag{14c}$$
$$\kappa_h = 0, \tag{14d}$$

therefore assuming both upflow $\chi$ and horizontal shear $\kappa_h$ to be constant. Notice that the upflow angle is set to 4.5°, which corresponds to the rotor uptilt.

The two models were then tested and compared in turbulent wind conditions. Three different combinations of inflow angles and shears (not included in the identification set) were considered, each using four different turbulent realizations, for a total of 12 tests performed at each given wind speed and turbulence intensity (TI). Figures 2 and 3 show, respectively, the mean (over 10 min and over all turbulent seeds) absolute error $\epsilon$ and standard deviation $\sigma$ as functions of wind speed, for two different levels of TI, equal to 5 % and 12 %. The results of the reference full model are shown using solid lines, while the ones of the rotationally symmetric formulation using dashed lines. The two formulations appear to be characterized by a very similar performance. Actually, notwithstanding its reduced identification set, the symmetric method obtains marginally better results. As expected, TI has a negative effect on the quality of the estimates. In addition, as already noticed in Bertelè et al. (2017), angle estimates appear to be less precise than shear estimates. Nonetheless, for 12 % TI at 15 m s$^{-1}$, the yaw misalignment mean error is about 2.5°. This appears to be a good result when compared to the typical accuracy of nacelle-mounted anemometers.

### 3.2 Verification with a scaled model in a wind tunnel

Next, the proposed formulation was tested using an aeroelastically scaled wind turbine operated in a boundary layer wind tunnel. The scaled model represents a three-bladed horizontal-axis wind turbine with a hub height of about

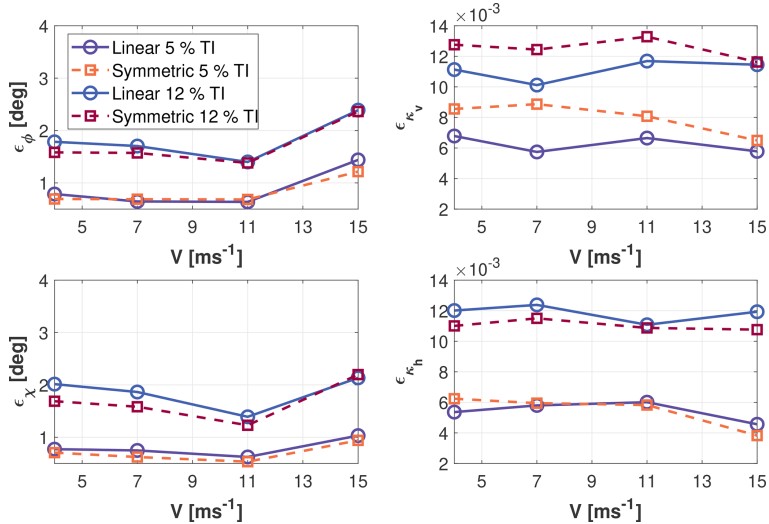

**Figure 2.** Mean absolute error $\epsilon$ of the four wind states vs. wind speed for 5 % and 12 % turbulence intensity (TI) levels. Nonsymmetric model: solid lines; symmetric model: dashed lines.

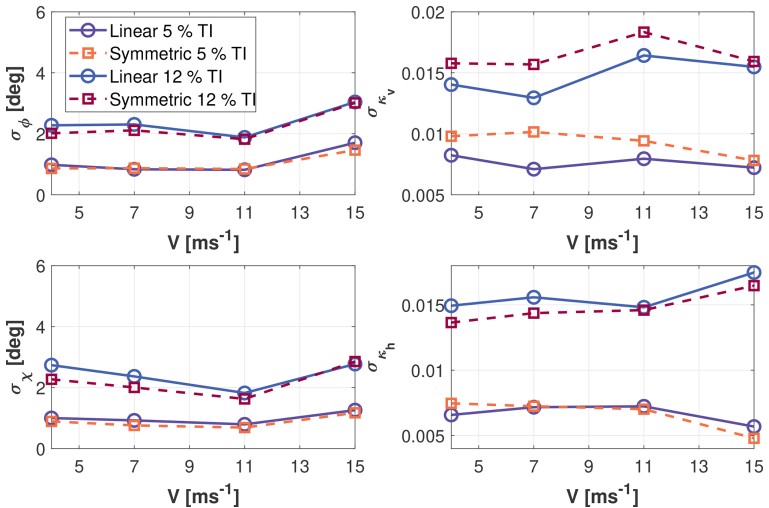

**Figure 3.** Standard deviation $\sigma$ of the four wind states vs. wind speed for 5 % and 12 % turbulence intensity (TI) levels. Nonsymmetric model: solid lines; symmetric model: dashed lines.

1.8 m, a rotor diameter of 2 m and a rated wind speed of 6 m s$^{-1}$ (Bottasso et al., 2014). The turbine design preserves the tip speed ratio, lock number CE1, and placement of the lowest tower and rotor nondimensional frequencies of the reference machine, resulting in a scaled model of realistic aeroelastic behavior (Bottasso et al., 2014). Each of the flexible scaled blades is equipped with strain gauges at the blade roots, which measure the flapwise and edgewise bending moments, while an optical incremental encoder is used to measure the blade azimuthal position.

Tests were performed in the boundary layer test section of the wind tunnel of Politecnico di Milano (Bottasso et al., 2014). Two different boundary layer conditions, characterized by different mean vertical shears and TI levels, were obtained by the use of suitable turbulence generators at the chamber inlet and roughness elements placed on the floor. Such inflow conditions were then accurately mapped over the rotor swept area with triple hot-wire probes, providing a reference mean inflow that can be considered the "ground truth". The lower turbulence condition was characterized by a TI of 3.8 % and a linear vertical shear of 0.03, while the higher turbulence case by a TI of 8.5 % and a linear vertical shear of 0.12.

For various wind speeds, several tests were performed for different combinations of yaw misalignment, vertical shear and upflow angle as reported in Table 1. Changes in mean vertical shear were obtained by changing the wind tunnel boundary layer conditions. Changes in mean misalignment

**Table 1.** Test matrix for the wind tunnel experiments. Symbol "×" marks the identification set; "∘" marks the validation set.

| | | \multicolumn{9}{c}{} |
|---|---|---|---|---|---|---|---|---|---|---|

Experiments conducted with an upflow angle $\chi = 6°$

| | | Misalignment angle $\phi$ (°) | | | | | | | | |
|---|---|---|---|---|---|---|---|---|---|---|
| Wind speed $V$ (m s$^{-1}$) | Vertical shear $\kappa_\mathrm{v}$ | 20 | 15 | 10 | 6 | 0 | −6 | −10 | −15 | −18 |
| 5 | 0.03 and 0.12 | × | | ∘ | | × | ∘ | | | ∘ |
| 5.5 | 0.03 and 0.12 | × | | ∘ | | × | ∘ | | | ∘ |
| 6 | 0.03 and 0.12 | | × | ∘ | ∘ | × | ∘ | ∘ | ∘ | |
| 7 | 0.03 and 0.12 | | × | ∘ | ∘ | × | ∘ | ∘ | ∘ | |
| 7.5 | 0.03 and 0.12 | | × | | ∘ | × | ∘ | | ∘ | |

Experiments conducted with upflow angles $\chi = 0$ and $12°$

| | | Misalignment angle $\phi$ (°) | | | | | | | | |
|---|---|---|---|---|---|---|---|---|---|---|
| Wind speed $V$ (m s$^{-1}$) | Vertical shear $\kappa_\mathrm{v}$ | 20 | 15 | 10 | 6 | 0 | −6 | −10 | −15 | −18 |
| 5 | 0.03 and 0.12 | ∘ | | ∘ | | ∘ | ∘ | | | ∘ |
| 5.5 | 0.03 and 0.12 | ∘ | | ∘ | | ∘ | ∘ | | | ∘ |
| 6 | 0.03 and 0.12 | | ∘ | ∘ | ∘ | ∘ | ∘ | ∘ | ∘ | |
| 7 | 0.03 and 0.12 | | ∘ | ∘ | ∘ | ∘ | ∘ | ∘ | ∘ | |
| 7.5 | 0.03 and 0.12 | | ∘ | | ∘ | ∘ | ∘ | | ∘ | |

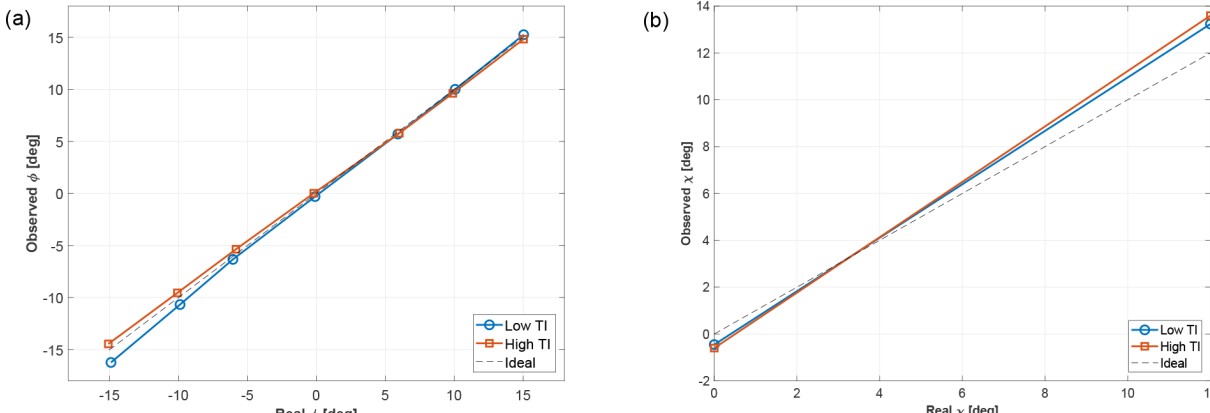

**Figure 4.** [CE2] This paper describes a new formulation for estimating the wind inflow at the rotor disk, based on measurements of the blade loads. The new method improves on previous formulations by exploiting the rotational symmetry of the problem. Experimental results obtained with an aeroelastically scaled model in a boundary layer wind tunnel are used for validating the proposed approach. Wind states observed for different steady inflow conditions: yaw misalignment $\phi$ at $\chi = 6°$ and $\kappa_\mathrm{h} = 0$ at a wind speed of $7\,\mathrm{m\,s^{-1}}$ (**a**), upflow angle $\chi$ at $\phi = 6°$ and $\kappa_\mathrm{h} = 0$ at a wind speed of $5.5\,\mathrm{m\,s^{-1}}$ (**b**).

angle were realized by yawing the turbine model with respect to the wind. To create different upflow angles, the wind turbine tower foot was installed on a tiltable ramp. By changing the ramp angle, the turbine can be pitched by ±6°. As the
5 rotor has an uptilt angle of 6° with respect to the tower, the use of the ramp allows one to obtain upflow angles between 0 and 12°. Finally, the horizontal shear for all tests can be considered null, as the flow in the wind tunnel is essentially uniform in the lateral direction.
10     A total number of 174 different conditions were tested. The entire set of experiments was then divided into two subsets. The first one was used for identifying the observer

model, and it contains two combinations of vertical shear and misalignment angle per wind speed, with an upflow of 6°; these test points are indicated with "×" symbols in Table 1. 15
The second subset was instead used for validating the observer performance. This second subset contains all the other experiments, indicated with "∘" symbols in Table 1. Notice that the second set of experiments correspond to upflow angles of 0 and 12°, values that are not contained in the iden- 20
tification set. This is possible thanks to the symmetry of the rotor: the information contained in the identification set on the effect of the misalignment angle is used to infer the ef-

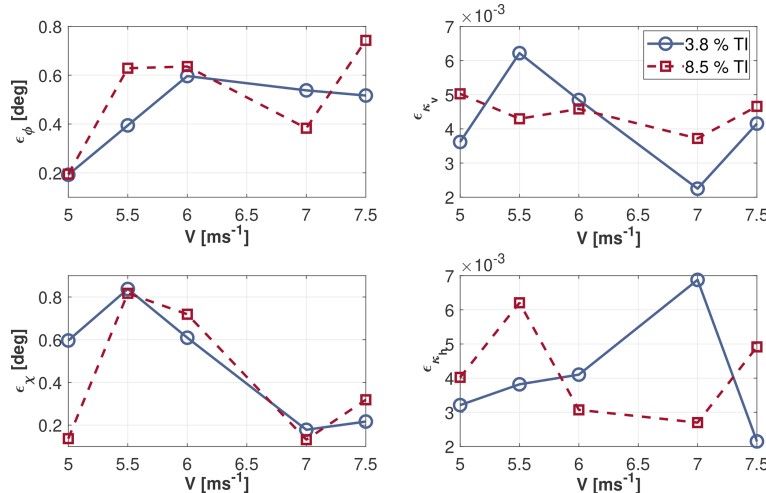

**Figure 5.** Mean absolute error $\epsilon$ of the four wind states vs. wind speed for 3.8 % and 8.5 % turbulence intensity (TI) levels.

fect of the upflow, although no operating points at different upflows are used during training.

To validate the performance of the observer, the machine response during each test was averaged over a time window of 180 s in order to estimate the corresponding mean inflow parameters. The length of the time window is dictated in this case not only by the need to average out turbulent fluctuations, but also by the dynamic characteristics of this particular closed-return wind tunnel. Figure 4 shows an excerpt of the results obtained at a wind speed of $7\,\mathrm{m\,s^{-1}}$ (Fig. 4a), which corresponds to the beginning of the full load region, and a speed of $5.5\,\mathrm{m\,s^{-1}}$ (Fig. 4b), which corresponds to the end of the partial load region. In each panel, the reference (true) wind parameter is shown on the $x$ axis, while the corresponding observed quantity is given on the $y$ axis. It follows that an ideal match would be represented by the bisector of the quadrant. The yaw misalignment estimation (Fig. 4a) appears to be quite accurate and has a maximum error of less than $1.3°$. Better accuracy can be achieved for high positive yaw angles; this is to be expected, since such conditions are included in the identification set (cf. Table 1). Even the upflow estimation (Fig. 4b) appears to be quite accurate, with a maximum error of about $1.5°$. Note that the accuracy in the upflow estimation validates the assumption of rotational symmetry of the parameters, as no upflow changes were present in the data set used for identifying the load-wind model (again, cf. Table 1). Indeed, the model coefficients related to this parameter were obtained using the symmetry conditions given by Eq. (12a–d TS12).

Finally, to better understand the performance of the observer, mean inflow parameters were estimated and compared to the respective ground truth for each test not included in the identification set. For each wind speed, such mean errors were averaged over the number of tests and reported in Fig. 5. Here again, results appear to be significantly accurate;

in fact, for both turbulence levels, a maximum mean error smaller than $1°$ is observed in the angle estimates, while the error in the shear estimates is less than $6 \times 10^{-3}$.

Comparing the experimental results with the numerical ones in the low TI cases (equal to 3.8 % and 5 %, respectively), one should notice that the mean estimation errors present the same range of accuracy, as one can appreciate by comparing Fig. 5 with Fig. 2. This can be considered an additional proof of the general applicability of the method, since these results were obtained with two different models applied to two very different machines, using numerical and experimental data sets.

## 4 Conclusions

Following the work presented in Cacciola et al. (2016a) and Bertelè et al. (2017), this paper has further developed and experimentally validated a method to estimate the inflow at the rotor disk. Specifically, a linear model was formulated to estimate four wind parameters: the vertical and horizontal shears, and the vertical and horizontal wind misalignments. Improving on the previous publications, the rotationally symmetric behavior of the rotor was exploited in order to simplify the model identification procedure, by reducing the number of necessary measured operating conditions.

The performance of the proposed rotationally symmetric model was tested both in simulation and with an aeroelastically scaled wind turbine model in a boundary layer wind tunnel. Results indicate no significant difference in the accuracy of the new rotationally symmetric formulation with respect to the original one, even if the number of tests required for identification is significantly decreased. The expected mean error in the angle estimation is less than 1 and $2.5°$ for low and high TI levels, respectively. An even higher accuracy can be obtained for the estimation of shears. More-

over, the experimental results are well in line with the ones obtained by numerical simulations.

**Data availability.** TS13

Please note the remarks at the end of the manuscript.

## Appendix A: Nomenclature

| | |
|---|---|
| $m$ | Generic blade moment |
| $\boldsymbol{m}$ | Vector of moment harmonics |
| $R$ | Rotor radius |
| $V$ | Wind speed |
| $\boldsymbol{V}$ | Wind vector |
| $\widetilde{v}$ | Non-dimensional horizontal cross flow at the hub |
| $\widetilde{w}$ | Non-dimensional vertical cross flow at the hub |
| $\varrho$ | Air density |
| $\phi$ | Yaw misalignment angle |
| $\chi$ | Upflow angle |
| $\kappa_{\mathrm{v}}$ | Vertical shear |
| $\kappa_{\mathrm{h}}$ | Horizontal shear |
| $\epsilon$ | Mean error |
| $\sigma$ | Standard deviation |
| $\boldsymbol{\theta}$ | Wind state vector |
| $(\cdot)^{\mathrm{T}}$ | Transpose |
| $(\cdot)_{\mathrm{E}}$ | Estimated quantity |
| $(\cdot)^{\mathrm{OP}}$ | Out-of-plane quantity |
| $(\cdot)^{\mathrm{IP}}$ | In-plane quantity |
| $(\cdot)_{1\mathrm{c}}$ | 1P cosine amplitude |
| $(\cdot)_{1\mathrm{s}}$ | 1P sine amplitude |
| BEM | Blade element momentum |
| Lidar CE3 | Light detection and ranging |
| TI | Turbulence intensity |
| 1P | Once per revolution |

**Author contributions.** TS14

**Competing interests.** The authors declare that they have no conflict of interest. TS15

**Acknowledgements.** This work has been partially supported by the CL-Windcon project, which receives funding from the European Union Horizon 2020 research and innovation program under grant agreement No. 727477.

This work was supported by the German Research Foundation (DFG) and the Technical University of Munich (TUM) in the framework of the Open Access Publishing Program.

Edited by: Sandrine Aubrun
Reviewed by: two anonymous referees

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

Please note the remarks at the end of the manuscript.

## Remarks from the language copy-editor

CE1    Please confirm the change.

CE2    "Hight" has been copy-edited to "High" in the legend of Fig. 4a. Please confirm the change.

CE3    "lidar" is a normal word and is in lower case throughout the text. Do you wish to remove it from this table?

## Remarks from the typesetter

TS1    Copernicus Publications collects the DOIs of data sets, videos, samples, model code, and other supplementary/underlying material or resources as well as additional outputs. These assets should be added to the reference list (author(s), title, DOI, and year) and properly cited in the article. If no DOI can be registered, assets can be linked through persistent URLs. This is not seen as best practice and the persistence of the URL must be secured.

TS2    Please confirm.

TS3    Dear author, please check throughout the text that all matrices are bold roman and all vectors are bold italic. Thanks.

TS4    Please confirm.

TS5    The composition of Figs. 2–5 has been adjusted to our standards.

TS6    Please confirm.

TS7    Please confirm.

TS8    Please confirm.

TS9    Please confirm.

TS10    Please confirm.

TS11    Please confirm equation above.

TS12    Please confirm.

TS13    Please provide a statement on how your underlying research data can be accessed. If the data are not publicly accessible, a detailed explanation of why this is the case is required. The best way to provide access to data is by depositing them (as well as related metadata) in reliable public data repositories, assigning digital object identifiers (DOIs), and properly citing data sets as individual contributions. Please indicate if different data sets are deposited in different repositories or if data from a third party were used. If no DOI is available, assets can be linked through persistent URLs to the data set itself (not to the repositories' home page). This is not seen as best practice and the persistence of the URL must be secured.

TS14    Copernicus Publications strongly recommends including the section "Author contributions".

TS15    Declaration of all potential conflicts of interest is required by us as this is an integral aspect of a transparent record of scientific work. If there are possible conflicts of interest, please state what competing interests are relevant to your work.

TS16    Please provide page range or article number.

TS17    Please confirm.

TS18    Please provide volume as well as page range or article number.

TS19    Is 1–8 the page range? Otherwise please provide page range.

TS20    Is 1–8 the page range? Otherwise please provide page range.

TS21    Is 1–8 the page range? Otherwise please provide page range.