# Peer review of "Wind inflow observation from load harmonics: wind tunnel validation of the rotationally symmetric formulation"

_Wind Energy Science, 2018_

## Referee Comment (RC1) · Anonymous Referee #1 · 19 Oct 2018

**Journal:** WES
**MS No.:** wes-2018-61
**MS Type:** Research articles
**Submission Date** 2018-09-14
**Date Due** 2018-11-01
**Title:** "Wind inflow observation from load harmonics: wind tunnel validation of the rotationally symmetric formulation"
**Author(s):** Marta Bertelè et al.

**General comments:**

This paper is in the continuity of the linear formulation black box model developed by Marta Bertelè et al. (2017) to predict the mean wind state inflows (shears and misalignments) from wind turbine loads. This paper presents an improvement of this formulation using the rotational symmetry of loads from this simple idea: effects on the loads of the vertical shear and misalignment are the same as those of the horizontal quantities shifted by pi/2. Then, this new formulation that enable the use of a reduced dataset for the model identification, was successfully validated in the simulation environment and in wind tunnel experiments.

The idea of the paper and how it was validated is worth for publication. The paper is well written and clear. I thus recommend the publication of the paper with minor corrections (see below).

**Minor corrections:**

P10: "the respective ground truth for each test not included in the identification set"
Can you please specify the test cases used here ?

The critical dataset to evaluate the rotational symmetry would be to use the +/- 6° of vertical misalignment (upflow angle) to identify the model, and then the use of a dataset with different yaw misalignments to test the performance of the observer. Did you tried that ?

---

## Referee Comment (RC2) · Anonymous Referee #2 · 19 Oct 2018

A useful paper, a good improvement and important tidying-up of the method as previously presented. Small comments:

Equation (7): there should be some explanation of where this comes from, otherwise it comes out of the blue.

End of Section 2.3: "since the model is linear": it would be useful if, somewhere in the paper, there was at least a comment on the appropriateness of a linear model for this, or some justification for assuming that non-linearities are not important here.

Section 3.1: "The model includes geometrically-exact blades", but does it include blade flexibility? Blade dynamic response and unsteady aerodynamic effects could poten-
tially cause problems for the method. Could, or should, the model be extended to allow for this, or can it be shown to be unimportant, even for the modern very long and flexible blades? Wind tunnel tests are useful up to a point, but are there any plans to test this on a large wind turbine?
* * *

---

## Author Comment (AC1) · 30 Nov 2018

**Reply to Reviewers**

We thank the reviewers for their detailed analysis and constructive inputs. A list of point-by-point replies to the reviewers' comments is reported in the following.

**Reviewer 1**

1) **[Reviewer]:** *P10: "the respective ground truth for each test not included in the identification set" Can you please specify the test cases used here?*
   *The critical dataset to evaluate the rotational symmetry would be to use the +/- 6° of vertical misalignment (upflow angle) to identify the model, and then the use of a dataset with different yaw misalignments to test the performance of the observer. Did you tried that ?*
   **[Authors]:** The entire set of experiments was divided into two subsets: the first one, reported in Eq. (11), was used for identifying the observer, while the second one for its validation. Hence, the second set was not included in the identification phase. To better describe this important point, the text in Section 3.2 was updated and a new table was added, reporting all experimental data points and their split into identification and validation subsets.

2) **[Reviewer]:** *The critical dataset to evaluate the rotational symmetry would be to use the +/- 6° of vertical misalignment (upflow angle) to identify the model, and then the use of a dataset with different yaw misalignments to test the performance of the observer. Did you tried that?*
   **[Authors]:** Not exactly, but we did something very similar and –we believe- more appropriate in the present context. In fact, we included in the identification set tests with constant upflow angle (6°) and changing yaw conditions, and then we used this model to estimate the inflow when different vertical misalignments were present (see Fig. 4, right). We believe that our approach is even more appropriate to validate the rotationally symmetric formulation of the observer. In fact, in a real environment, it is more probable to experience significant changes of yaw misalignment than variations of the upflow angle (indeed, one might possibly intentionally misalign the machine, while the same is not possible for the upflow). Hence, having a possible field application in view, it is better to consider the upflow as fixed to a constant value for the identification phase and, later, let it vary in the verification step. The new text and table of Section should help make this point clearer in the new version of the manuscript.

**Reviewer 2**

1) **[Reviewer]:** *Equation (7): there should be some explanation of where this comes from, otherwise it comes out of the blue.*
   **[Authors]:** The text was improved to better explain how this result is obtained.

2) **[Reviewer]:** *End of Section 2.3: "since the model is linear": it would be useful if, somewhere in the paper, there was at least a comment on the appropriateness of a linear model for this, or some justification for assuming that non-linearities are not important here.*
   **[Authors]:** Indeed a non-linear formulation is also possible, as described in the third reference of the paper, i.e. Bertelè, M., Bottasso, C.L. and Cacciola, S.: Wind inflow observation from load harmonics, Wind Energ. Sci., doi: 10.5194/wes-2017-23, 2017. A sentence was added at the beginning of Section 2.2, where the linear model is introduced, to explain this fact.

3) **[Reviewer]:** *Section 3.1: "The model includes geometrically-exact blades", but does it include blade flexibility? Blade dynamic response and unsteady aerodynamic effects could potentially cause problems for the method. Could, or should, the model be extended to allow for this, or can it be shown to be unimportant, even for the modern very long and flexible blades?*
**[Authors]:** Yes, "geometrically exact" refers to a formulation for flexible bodies where kinematical non-linearities due to finite rotations and large displacements are treated without approximations. The model also accounts for unsteady aerodynamic effects, tip losses and dynamic stall. The text has been improved to clarify these points.

4) **[Reviewer]:** *Wind tunnel tests are useful up to a point, but are there any plans to test this on a large wind turbine?*
**[Authors]:** We certainly agree. We just started working on a new funded research program, where we are scheduled to demonstrate the formulation in the field using a multi-MW wind turbine. Any findings will be duly reported in future publications.

We have taken the opportunity to make several small editorial changes to the text, in order to improve readability. Additionally, we have also slightly changed the description of the formulation and we have updated some of the plots with new results, obtained with a larger number of seeds. A revised version of the manuscript is attached to the present reply, with the main changes highlighted in red.

Best regards.
The authors